# Diversity of CRISPR-Cas Systems Identified in Urological *Escherichia coli* Strains

**DOI:** 10.3390/microorganisms13122846

**Published:** 2025-12-15

**Authors:** Pavel V. Slukin, Mikhail V. Fursov, Daniil V. Volkov, Angelika A. Sizova, Konstantin V. Detushev, Ivan A. Dyatlov, Nadezhda K. Fursova

**Affiliations:** State Research Center for Applied Microbiology and Biotechnology, 24 Territory “Kvartal A”, 142279 Obolensk, Russia; xopgi@yandex.ru (P.V.S.); mikhail.fursov88@gmail.com (M.V.F.); volkov@obolensk.org (D.V.V.); sizova@obolensk.org (A.A.S.); detushevkv@obolensk.org (K.V.D.); dyatlov@obolensk.org (I.A.D.)

**Keywords:** CRISPR-Cas systems, uropathogenic *Escherichia coli* (UPEC), spacer sequences, type I-E, type I-F, virulence genes, antimicrobial resistance genes, whole-genome sequencing, genetic diversity

## Abstract

Type I-E and I-F CRISPR-Cas systems were identified in 237 *E. coli* strains isolated from patients with urinary tract infections (UTIs) between 2004 and 2019. The strains were classified into nine distinct groups (I–IX) based on the presence or absence of *cas* genes and repeat regions (RRs). Within the type I-E systems, two sequence variants were identified, distinguished by polymorphisms in the *casB*, *cas3*, *cas7*, *cas5*, and *cas6* genes. The direct repeats (DRs) also differed, with I-E-associated RRs ranging from 26 to 32 bp and I-F-associated RRs consistently being 28 bp. We identified 762 unique spacers (29–35 bp in length) across the strain collection, while the number of spacers per strain varied from 1 to 47, and potential DNA targets were determined for 65 spacers, targeting 38 bacteriophage genomes, 19 plasmids, and 8 *cas* genes of the I-F type CRISPR-Cas system. Multilocus sequence typing (MLST) revealed 68 sequence types and 24 clonal complexes (CCs), with the most prevalent being ST131, CC10, CC69, CC405, CC14, CC38, CC73, and CC648. Significant correlations were observed between specific phylogroups/CCs, the type of CRISPR-Cas system present, and distinct profiles of virulence and antibiotic resistance genes.

## 1. Introduction

Uropathogenic *E. coli* (UPEC) is the primary causative agent of urinary tract infections (UTIs), and is responsible for up to 90% of community-acquired and 50% of hospital-acquired cases [1]. The pathogenicity of UPEC is driven by a diverse arsenal of virulence genes involved in adherence, invasion, and iron acquisition. Furthermore, whole-genome sequencing (WGS) data confirmed the presence of a wide variety of resistance genes in *E. coli* genomes of this pathotype [2].

The CRISPR-Cas system comprises Clustered Regularly Interspaced Short Palindromic Repeats (CRISPR) and CRISPR-associated (Cas) proteins, and provides bacteria with an adaptive immune defense against foreign DNA. It was first identified in *Escherichia coli* and studied in detail in the *E. coli* strain K-12. This foundational work established the functions of individual Cas proteins, elucidated the molecular mechanisms governing system assembly and activity, and identified key regulatory elements, including associated promoters and repressors [3,4]. In the past, the system was reported to be present in 45–50% of bacteria [5]. To date, additional functions of CRISPR-Cas systems have been identified, including the regulation of cryptic prophages [6,7].

Three CRISPR regions have been identified in the *E. coli* K-12 strain: *iap–cysH*, *clpA–infA*, and *ygcE–ygcF*. The first two regions contain *cas* genes, whereas the third comprises only RR sequences [8,9]. The *iap–cysH* and *clpA–infA* regions encode distinct sets of *cas* genes corresponding to the I-E (Ecoli or CASS2) and I-F (Ypest or CASS3) type systems, respectively [10,11]. Moreover, the type I-E system of *E. coli* K-12 has been shown to be similar to those of *Salmonella enterica* and *Klebsiella pneumoniae*, whereas the type I-F system shares similarity with that of *Yersinia pestis* [11,12]. It has been suggested that the *iap-cysH* and *ygcE-ygcF* regions are more associated with protection against phage DNA, and the *clpA-infA* region with protection against plasmid DNA, but another study found the opposite relationship [13,14].

The nomenclature of CRISPR-Cas systems, particularly that of Cas proteins, has not been standardized. For example, the CasA protein is also referred to as Cse1 or Cas8e, and CasB as Cse2. Furthermore, there is no consistent nomenclature for RRs [9]. The type I-E CRISPR-Cas systems comprise eight Cas proteins: Cas3, CasA, CasB, Cas7, Cas5, Cas6, Cas1, and Cas2. Cas1 and Cas2 are involved in the processes of spacer acquisition and adaptation, whereas CasA, CasB, Cas7, Cas5, and Cas6 mediate CRISPR RNA processing and the assembly of the Cascade effector complex, which recognizes foreign nucleic acids. Finally, Cas3 cleaves the target DNA through its nuclease activity [15].

CRISPR arrays of CRISPR-Cas systems consist of direct repeats (DRs) and intervening spacers homologous to fragments of foreign DNA. Together, these elements form a repeat region (RR) [16,17]. Analysis of spacer targets across a large number of *E. coli* strains revealed that at least 59% of spacers are homologous to bacteriophage DNA [9]. Notably, spacers within *E. coli* CRISPR arrays frequently target DNA sequences derived from bacteriophage P7 and epidemiologically relevant plasmids such as pO111 and pO157 [9,18].

Although recent research on CRISPR-Cas systems has primarily focused on their applications in laboratory, medical, and industrial contexts, there is growing interest in elucidating their roles within natural populations of both pathogenic and nonpathogenic *E. coli*. For instance, associations between CRISPR-Cas systems and antimicrobial resistance (AMR) have been demonstrated in avian pathogenic *E. coli*. Moreover, correlations between CRISPR-Cas systems, phylogroup classification, and sequence type have been observed in *E. coli* strains isolated from the birth canal of healthy women during the postpartum period. In addition, an association between CRISPR-Cas systems and virulence genes has been reported for UPEC [14,19,20,21].

This study focuses on the analysis of CRISPR-Cas systems in *E. coli* strains isolated from the urine of patients presenting with symptoms of urinary tract infection (UTI), with particular emphasis on their correlations with virulence factors, antibiotic resistance, and genetic lineage affiliation.

## 2. Materials and Methods

### 2.1. Strains and Growth Conditions

A total of 237 *E. coli* strains were obtained from the State Collection of Pathogenic Microorganisms “SCPM-Obolensk” and were isolated between 2004 and 2019 in the Central Region of the Russian Federation (Appendix A); bacteria were cultured on LB agar medium (Thermo Fisher Scientific, Waltham, MA, USA).

### 2.2. DNA Isolation and PCR Amplification

Genomic DNA for PCR was extracted by alkaline lysis. PCR amplification was carried out using a Mini Amp Plus instrument (Applied Biosystems Inc., Woburn, MA, USA) with the DreamTaq Green PCR Master Mix kit (Thermo Fisher Scientific, Waltham, MA, USA). Detection of specific genes and assignment to phylogenetic groups were performed as described by Clermont et al., 2015 [22].

### 2.3. Whole-Genome Sequencing

WGS was carried out on the Illumina MiSeq platform using the Nextera DNA Library Preparation Kit and MiSeq Reagent Kits v3 (Illumina, San Diego, CA, USA) following the manufacturer’s instructions. The resulting single-end reads were assembled into contigs using Unicycler v0.5.1 software (https://github.com/rrwick/Unicycler/releases/tag/v0.5.1, accessed on 14 December 2025). The whole-genome sequences of 55 strains have been deposited in the GenBank database under BioProject accession number PRJNA269675 (Appendix A).

### 2.4. Genomic Analysis

Quality control of genome assemblies was performed using the genome assembly evaluation tool QUAST v5.3.0 (https://github.com/ablab/quast, accessed on 9 December 2025) and the genome quality assessment tool CheckM2 v1.1.0 (https://github.com/chklovski/CheckM2, accessed on 9 December 2025).

WGS were analyzed using the Center for Genomic Epidemiology web services MLST 2.0, ResFinder 4.1, and PlasmidFinder [23,24,25] (http://www.genomicepidemiology.org/, accessed on 9 October 2025). Virulence genes of the UPEC pathotype strains were identified using the BLAST web service (http://blast.ncbi.nlm.nih.gov/Blast.cgi, accessed on 9 October 2025), and reference sequences of *E. coli* virulence genes were selected from four functional groups, as follows: (i) Adhesins: *fimH* (GenBank AJ225176: 1340–2242), *yfcV* (CP015085.1: 1243313–1250325), *papGII* (AY212279.1), *afaA* (X76688.1: 3471–3776), *papGIII* (AY212281.1), *sfaS* (X16664: 22118–22609), and *focG* (DQ301498: 6744–7247). (ii) Toxins: *hlyA* (M10133: 1320–4391), *usp* (AB027193), *cnf1* (X70670), and *vat* (NZ_JSGK01000039: 85437–89567). (iii) Siderophores: *fyuA* (AM236324: 43579–45600), *chuA* (AP017620.1: 3851222–3853204), *iutA* (AY553855: 5766–7964), and *iroN* (X16664: 25805–27982). (iv) Protectins: *traT* (AY684127.1), *ompT* (KP657558.1), and *kpsMT* (AP024112.1: 870921–872368).

The primary identification of CRISPR-Cas systems was carried out using the CRISPR-Cas++ web service (https://crisprcas.i2bc.paris-saclay.fr/, accessed on 9 October 2025), while refinement analysis was performed using the BLAST web service (http://blast.ncbi.nlm.nih.gov/Blast.cgi, accessed on 9 October 2025). Complete genomes of *E. coli* strains K-12 substr. MG1655 (U00096.3), B-8552 (JAHWDZ000000000.1), and B-8431 (SERV00000000.1) were used as reference sequences.

Phylogenetic analysis of whole-genome sequences was performed using Snippy version 4.6.0 with default settings. The resulting phylogenetic tree was visualized using the iTOL web resource (https://itol.embl.de/, accessed on 9 October 2025).

### 2.5. Statistical Methods

Statistical analysis was performed using GraphPad Prism 8.0.1. One-way ANOVA with post hoc Tukey’s multiple comparisons test was used, and significance was assumed at *p* < 0.05.

## 3. Results

### 3.1. General Characteristics of the Strains

A total of 237 *E. coli* strains were obtained from the State Collection of Pathogenic Microorganisms “SCPM-Obolensk”. These strains were isolated between 2004 and 2019 from ten medical hospitals across Russia, and originated from the urine of patients with the following clinical diagnoses: urinary tract infection of unknown localization (*n* = 192), chronic cystitis (*n* = 25), asymptomatic bacteriuria (*n* = 6), pyelonephritis (*n* = 4), chronic pyelonephritis (*n* = 4), cystitis (*n* = 2), urolithiasis (*n* = 2), gestational pyelonephritis (*n* = 1), and overactive bladder (*n* = 1).

The strains were attributed to seven phylogenetic groups of *E. coli*: B2 (54%), A (16%), D (15%), B1 (9%), F (2%), E (2%), and C (1%). Among 231 strains, 70 sequence types (STs) were identified according to the Warwick University scheme, belonging to 24 clonal complexes (CCs). CC131 was predominant (*n* = 85, 35%), while other common CCs included CC10 (ST10, ST167, ST617; 8%), CC69 (ST69; 5%), CC405 (ST405; 4%), CC14 (ST14, ST1193, ST1858; 3%), CC38 (ST38, ST315; 3%), CC73 (ST73, ST1154; 3%), and CC648 (ST648, ST3177; 3%). Sixteen rarer clonal complexes (CC23, CC101, CC155, CC59, CC95, CC350, CC12, CC46, CC86, CC156, CC165, CC168, CC349, CC354, CC394, and CC469) were each represented by 1–3 strains (Appendix A).

Genetic determinants of antibiotic resistance were identified in 203 *E. coli* strains, with resistance to beta-lactams (80%), fluoroquinolones (70%), aminoglycosides (63%), sulfonamides (63%), tetracyclines (57%), phenicols (42%), macrolides (3%), polymyxins (2%), fosfomycins (1%), and ansamycins (1%) observed. Virulence genes associated with the uropathogenic *E. coli* (UPEC) pathotype were detected in the strain genomes, including *fimH* (95%), *fyuA* (76%), *chuA* (68%), *traT* (68%), *iutA* (66%), *ompT* (64%), *yfcV* (55%), *usp* (53%), *kpsMTII* (46%), *hlyA* (25%), *iroN* (22%), *cnf1* (21%), *papGII* (17%), *vat* (14%), *afa* (12%), *papGIII* (10%), *sfaS* (7%), *focG* (4%), and *kpsMTIII* (3%) (Appendix A).

### 3.2. Identification and Characterization of CRISPR-Cas Systems

Whole-genome sequences of 237 *E. coli* strains were analyzed for the presence of CRISPR-Cas system components. In total, 127 strains carried CRISPR-Cas systems, including 110 assigned to type I-E and 17 to type I-F. The remaining 110 strains lacked CRISPR-Cas systems (i.e., did not possess genes encoding Cas proteins), although 108 of these contained RRs (Figure 1).

Comparative analysis of whole-genome sequences identified three CRISPR-Cas-associated regions in the genomes of uropathogenic *E. coli* strains, which were analogous to those in the reference strain *E. coli* K-12 substr. MG1655. Region A is located between the *clpA* gene, a component of the Clp chaperone–protease operon, and the *infA* gene encoding translation initiation factor IF-1. Region B was found between the *iap* gene encoding alkaline phosphatase isoenzyme aminopeptidase and the *cysH* gene for phosphoadenosine-phosphosulfate reductase. Region C was located between the sucrose kinase gene *ygcE* and *queE*, which encodes 7-carboxy-7-deazaguanine synthase (Figure 1).

Region A contains a single RR1 in a large proportion (48%) of the studied strains (Groups I, II, VI, and IX). In 7% of strains, this region contains two distinct RRs, designated RR1 and RR2, which are separated by the type I-F *cas* gene cluster (Groups VII and VIII). A small subset of strains (2%) lacks any CRISPR-Cas structures in Region A (Groups III, IV, and V). Furthermore, the *serW* tRNA gene was identified between the *cas* gene cluster and RR2 in Groups VII and VIII. This gene is also present in Region A of all other strains, including those lacking other CRISPR-Cas components (Figure 1).

Region B contains the RR3 locus and a set of the type I-E *cas* genes in 46% of strains (Groups I–IV). In the remaining strains (Groups V–IX), this region lacks CRISPR-Cas components. Additionally, Region B harbors genes of the Hok toxin–antitoxin system. In 90% of strains, this system is represented by both the *hok* toxin gene and the *sokX* antitoxin gene located adjacent to *cysH*, while in the remaining strains, only the *sokX* gene is present, similar to the reference *E. coli* K-12 substr. MG1655 strain (Figure 1).

Region C includes the RR4 locus in 46% of strains (Groups I–III). In 47% of strains (Groups IV–VII), this region carries a set of carbohydrate metabolism genes, including *kdgK* (encoding aminoimidazole riboside kinase), *scrY* (carbohydrate porin), *sacX* (subunit IIBC of the sucrose transporter), *scrB* (sucrose-6-phosphate hydrolase), and *lacI* (a transcriptional regulator). In the remaining strains (Groups VIII–IX), Region C lacks any genes (Figure 1).

### 3.3. Types of CRISPR-Cas Systems

Two types of CRISPR-Cas systems were identified among the studied uropathogenic *E. coli* strains, namely type I-E (*n* = 110), located in genomic Region B, and type I-F (*n* = 17), identified in genomic Region A (Figure 1).

The gene order and composition of the type I-E CRISPR-Cas systems (Groups I–IV) were as follows: *cas3*-*casA*-*casB*-*cas7*-*cas5*-*cas6*-*cas1*-*cas2*. However, based on sequence divergence, two distinct variants were identified. The first variant (var. K-12), identical to the reference strain *E. coli* K-12 substr. MG1655 was found in 11% of strains (Group I), while the second variant (var. B-8552), identified in Groups II–IV, was characterized by substantial differences in the *cas3*, *cas7*, *cas5*, and *cas6* genes. These genes shared no significant nucleotide similarity with the var. K-12, while their encoded proteins exhibited 29–31% amino acid identity (with coverage > 50%). In contrast, the *casA*, *cas1*, and *cas2* genes and their corresponding protein products showed high sequence similarity to those in var. K-12. The *casB* gene and its protein product were generally unique to each variant set (Appendix A, Table 1).

The order and identity of *cas* genes in type I-F CRISPR-Cas systems were as follows: *cas1f*-*cas3f*-*csy1*-*csy2*-*csy3*-*cas6f* (*n* = 17). Although the strains carrying this type of CRISPR-Cas system are divided into two groups (Groups VII and VIII), the distinction between them pertains solely to the genetic content of Region C, while the structure and composition of the I-F CRISPR-Cas system itself are identical in both (Figure 1).

### 3.4. Analysis of the Repeat Regions (RRs)

Our analysis revealed the following distribution of RRs: in type I-F CRISPR-Cas systems, the combination of RR1 and RR2 was exclusive; in type I-E systems, several combinations were identified (RR1 together with RR3 and RR4, RR3 and RR4 together, or RR3 alone). Additionally, RR1 was found as a standalone repeat locus in strains that lacked any associated *cas* genes (Figure 1).

#### 3.4.1. Direct Repeats (DRs)

Our analysis revealed that the DRs of RR1 and RR2, located within genomic Region A, consisted of identical 28 bp sequences (tttctaagctgcctgtacggcagtgaac) in the majority of strains (*n* = 232). In two strains, an additional cytosine was present at the right end of the sequence (Appendix A).

The DRs within RR3 and RR4 (located in genomic Regions B and C, respectively) are 26–32 nucleotides in length. These sequences share a conserved 24-nucleotide core (tttatccccgctggcgcggggaac), which is flanked by 2–4 variable nucleotides on either end. Furthermore, some strains possess point mutations within this core sequence. The only exception to this overall pattern was a single 29 bp DR identified in one strain (Appendix A, Figure 2).

Thus, we showed the differences in the DR sequences of RR1 and RR2 (CRISPR-Cas I-F) compared to the DR sequences of RR3 and RR4 (CRISPR-Cas I-E).

#### 3.4.2. Spacer Sets

A total of 762 unique spacers were identified across the studied urological *E. coli* strains. Spacer lengths ranged from 29 to 35 bp, with 32 bp spacers being the most prevalent (80%). Some spacers from different strains exhibited high sequence similarity, differing only by single nucleotide substitutions, and were therefore grouped into sequence clusters. The number of spacers per strain varied from 1 to 47, with two strains lacking spacers entirely. The most widely distributed spacers were sp876, sp665, and sp608, found in 41%, 23%, and 17% of strains, respectively. Notably, sp608 was present as two adjacent copies in six strains. The majority of spacers (42%) were unique, each identified in only a single strain (Appendix A).

Differences in the number of spacers among the RRs were observed: RR1 contained 1–14 spacers, RR2 2-33, RR3 2-30, and RR4 2-27. In total, 59 distinct spacers were identified in RR1, 50 in RR2, 355 in RR3, and 340 in RR4. The most prevalent spacers in RR1 were sp876 (*n* = 97), sp665 (*n* = 54), and sp608 (*n* = 46); in RR2, sp308 (*n* = 13), sp3 (*n* = 11), and sp11 (n = 11); in RR3, sp81 (*n* = 36), sp42 (*n* = 28), and sp250 (*n* = 17); and in RR4, sp151 (*n* = 26), sp398 (*n* = 24), sp49 (*n* = 23), and sp122 (*n* = 23). Notably, the majority of spacers (95%) were present in only one RR, 5% in two RRs, and only two spacers occurred in three RRs (Appendix A).

BLAST analysis identified potential DNA targets for 65 spacers in the GenBank database. These targets were located in bacteriophage genomes (*n* = 38), plasmids (*n* = 19), and Cas protein coding sequences (*n* = 8). For instance, 28 spacers matched different regions of the *Escherichia* phage vB_EcoM-705R4 genome (GenBank ON470624.1), seven matched the *Escherichia* phage PhiR41_1 genome (GenBank PV340561.1), and three matched the *Enterobacteria* phage P7 genome (GenBank NC_050152.1) (Appendix A, Figure 3).

Target DNAs for nineteen spacers were identified on plasmids, including targets for twelve spacers (sp2, sp7, sp187, sp271, sp293, sp318, sp370, sp425, sp529, sp574, sp679, and sp846) on the pGF54-C cryptic plasmid of an unidentified Inc group (GenBank CP172162.1). Seven spacers (sp40, sp270, sp347, sp533, sp536, sp623, and sp874) matching sequences on the p1-S1-IND-01-A plasmid of IncFIB group (GenBank CP145658.1) carrying both antimicrobial resistance genes (*bla*_CTX-M-15_, *qnrS1*, *sul1*, *aadA5*, and *dfrA17*) and virulence gene *ibeC* providing the overcoming of the blood–brain barrier and subsequent penetration into the endothelial cells of the brain. Six spacers (sp228, sp246, sp289, sp411, sp688, and sp714) matching sequences on the pEc2-51408 cryptic plasmid of IncFIB group (GenBank CP104116.1). Six spacers (sp40, sp347, sp533, sp536, sp623, and sp874) matching sequences on the pO157 hybrid plasmid of IncFIA:FIB:FII group (GenBank NZ_ABHM02000004.1) carrying a set of virulence genes (*hlyABCD*, *eptO*, *gspCDEFGHIJKL*, *stcE*, *ibeC*, and *espP*) specific for two pathotypes, Enterohemorrhagic *E. coli* (EHEC) and Enteroinvasive- and Meningitis-associated *E. coli* (EIEC/MNEC). Three spacers (sp536, sp623, and sp874) matching sequences on the p666 plasmid of the IncFII group (GenBank FN649417.1) carrying genes *eltAB* and *estIa* encoding virulence factors characteristic of enterotoxigenic *E. coli* (ETEC). Two spacers (sp270 and sp533) matching sequences on the pO83_CORR hybrid plasmid of IncFIB/IncFIC/IncQI group (GenBank NC_017659.1) carrying a huge set of antimicrobial resistance genes (*bla*_TEM-1_, *sul1*, *sul2*, *aadA1*, *dfrA1*, *tetA*, *catA1*, *aph(3″)-Ib*, *aph(6)-Id*, and *mphB*), as well as virulence genes (*iroBCDEN*, *iutA*, *iucABCD*, and *sitABCD*), which are a signature of Extraintestinal Pathogenic *E. coli* (ExPEC) with high invasive potential (Appendix A, Figure 4).

Furthermore, the target DNAs of eight spacers (sp28, sp487, sp491, sp608, sp665, sp673, sp875, and sp876) located in the RR1 of the studied *E. coli* strains of Groups I, II, VI, and IX (91% of strains) were identified in the sequences of the type I-F *cas* genes. Two spacers matched the *cas1f* gene, five spacers matched the *cas3f* gene, and one spacer matched the *csy3* gene (Appendix A, Figure 5).

### 3.5. E. coli Phylogenetic Groups and CRISPR-Cas System Prevalence

Phylogenetic analysis of uropathogenic *E. coli* genomes revealed three major clusters. Strains from CC131 formed a distinct, separate cluster, while the remaining strains were distributed between two heterogeneous groups comprising multiple clonal complexes and sequence types. A clear correlation emerged between phylogenetic clustering and the presence of specific CRISPR-Cas systems; for example, all strains belonging to the predominant clonal complex, CC131, lacked *cas* genes, although RR1 sequences were present in their genomes (Group VI). Conversely, every strain within the large cluster at the bottom of the phylogenetic tree (Figure 6) possessed a type I-E CRISPR-Cas system (Groups I–IV). Similarly, all genomes encoding type I-F systems (Groups VII, VIII) were exclusively concentrated within a separate, distinct cluster. No isolates with or without CRISPR-Cas systems were found that belonged to the same ST in the studied collection (Figure 6).

### 3.6. Correlation Between CRISPR-Cas System Type and Virulence Genes and AMR Genes

This analysis was performed on three strain cohorts: those carrying the type I-E CRISPR-Cas system, those harboring the type I-F system, and those lacking these systems. Strains in the I-E cohort carried significantly fewer virulence determinants overall compared to the other two cohorts, and this trend was most pronounced for specific genes, including those encoding the adhesins SfaS, FocG, and YfcV; the siderophores IutA, FyuA, and ChuA; the toxins Cnf1, HlyA, Usp, and Vat; and the protectins OmpT and KpsMTII (Figure 7a).

The data demonstrate that a high abundance of virulence genes in UPEC strains is positively correlated with either the presence of a type I-F CRISPR-Cas system or the complete absence of any CRISPR-Cas system (Figure 7b).

An analysis of the AMR genetic determinants revealed that strains in the type I-F cohort carried significantly fewer AMR genes compared to the other two cohorts. These differences were most pronounced for genes conferring resistance to beta-lactams, aminoglycosides, sulfonamides, tetracyclines, fluoroquinolones, macrolides, and polymyxins (Figure 8a). Overall, these data demonstrate a significant positive correlation between AMR gene prevalence in UPEC strains and either the presence of type I-E CRISPR-Cas systems or the complete absence of CRISPR-Cas systems (Figure 8b).

Moreover, it was shown that the studied strains carrying the type I-E CRISPR-Cas system and those lacking a CRISPR-Cas system carried a statistically higher number of plasmid replicons per strain compared to strains containing the type I-F CRISPR-Cas system. The most prevalent replicon was IncF, followed by pCol, IncI, IncX, and IncY (Figure 9, Appendix A).

## 4. Discussion

In this study, we analyzed the whole-genome sequences of 237 uropathogenic *E. coli* (UPEC) strains collected from patients at ten medical hospitals across the Russian Federation between 2004 and 2019. Our analysis assigned the strains to seven phylogenetic groups (A, B1, B2, C, D, E, and F), 68 sequence types, and 24 clonal complexes (CCs). The most prevalent lineages included CC131, CC10, CC69, CC405, CC14, CC38, CC73, and CC648. The strains harbored a diverse repertoire of virulence genes encoding adhesins (*fimH*, *sfaS*, *focG*, *papGII*, *papGIII*, *afaA*, *yfcV*), siderophores (*iroN*, *iutA*, *fyuA*, *chuA*), toxins (*cnf1*, *hlyA*, *usp*, *vat*), and protectins (*ompT*, *traT*, *kpsMTII*, *kpsMTIII*). Antimicrobial resistance (AMR) profiling revealed that over 50% of strains carried genes conferring resistance to beta-lactams, fluoroquinolones, aminoglycosides, sulfonamides, tetracyclines, and phenicols, while fewer than 5% carried resistance determinants for macrolides, polymyxins, fosfomycins, and ansamycins. These genomic characteristics align with global epidemiological patterns of UPEC dissemination, virulence gene representation, and antibiotic resistance prevalence [2,26,27,28].

A significant characteristic of our collection was the presence of CRISPR-Cas systems in 53% of the strains. This prevalence is substantially higher than the 18% reported in a previous study from Greece [2]. The type I-E system was the most prevalent, identified in 46% of strains, while the type I-F system was less common, present in 7%. Phylogenetic analysis revealed that strains carrying type I-E systems were distributed across groups A, B1, B2, C, D, and F. In contrast, strains with type I-F systems or those lacking CRISPR-Cas systems predominantly belonged to phylogroup B2, consistent with published data [14].

The genomic architecture of CRISPR-Cas systems in the studied strains is similar to that of the reference strain *E. coli* K-12 substr. MG1655, with components distributed across three designated regions (A, B, and C) [29]. Based on the presence or absence of *cas* genes, the composition of repeat regions (RR1-RR4), and additional genetic variations within these regions, the strains were categorized into nine distinct groups. These include strains carrying type I-E systems (Groups I–IV), type I-F systems (Groups VII–VIII), strains lacking *cas* genes but containing RRs (Groups VI, IX), and strains deficient in both *cas* genes and RRs (Group V) (Figure 1).

A key finding of our study is the identification of previously uncharacterized sequence divergence in the *cas3*, *cas7*, *cas5*, and *cas6* genes within type I-E CRISPR-Cas systems. The first variant (var. K-12), identical to the system in the reference strain *E. coli* K-12 substr. MG1655 was identified in 11% of strains (Group I). In contrast, a second, distinct variant (designated var. B-8552) was more prevalent, found in 35% of strains (Groups II-IV). Presumably, the *cas* genes of *E. coli* ST69 strains, which formed a separate clade among the *cas* genes of type I-E, described in a recent study, could be classified as having different variants of *cas* gene sequences [14].

Moreover, we identified incomplete *cas* operons (containing three, five, or six genes) in seven strains, consistent with previous reports [14]. These truncated operons consistently included *cas1*, *cas2*, and *cas3* genes, suggesting potentially diminished CRISPR-Cas activity. The occurrence of such deletions likely results from recombination events during bacterial evolution [2]. The analysis systematically defines the distribution and structural variation of CRISPR-Cas systems within a collection of uropathogenic *E. coli* strains. By categorizing strains into specific groups based on the presence, type, and genomic organization of these systems. This detailed classification serves as a foundational framework for future studies investigating potential links between specific CRISPR-Cas architectures and bacterial phenotypes.

Further supporting the horizontal transfer of *cas* genes, we observed that closely related strains harbored different *cas* operon variants. Among ST10 strains, seven carried the var. The K-12 system possessed the var. B-8552 variant. Similarly, ST648 strains included four strains with var. K-12 and one strain with var. B-8552 (Appendix A).

Additional evidence for horizontal transfer comes from the identification of CRISPR-Cas components on plasmids in public databases, including the following: an unnamed 69,879 bp plasmid (CP089250.1) from an *E. coli* strain collected in South Korea, 2021; plasmid pIOMTU792 (LC542972.1) from a Nepalese human isolate, collected in 2020; and IncA/C2 plasmid p24C171-1 (LC501671.1), from a Japanese broiler isolate, collected in 2012.

Our analysis identified two distinct groups of direct repeats (DRs). The first group consisted of 28 bp DRs with the core sequence tttctaagctgcctgtacggcagtgaac located in genomic Region A (RR1 and RR2). The second group comprised 24 bp DRs with the core sequence tttatccccgctggcgcggggaac found in Regions B and C (RR3 and RR4). This observed polymorphism in CRISPR loci, particularly in DR sequences, has been previously documented in enterobacterial genomes [11,12].

We identified 762 unique spacers across the uropathogenic *E. coli* strain collection. Spacer counts per strain ranged from 1 to 47, with lengths varying between 29 and 35 bp. BLAST analysis revealed potential DNA targets for 65 spacers, including 38 bacteriophage genomes, 19 plasmids, and 8 *cas* genes. Other studies have highlighted a significant number of spacers exhibiting no significant similarity with sequences submitted to the GenBank database [2]. Interestingly, in *E. coli*, there appears to be a correlation between the distribution of the spacer number and pathogenic traits [30].

Our analysis confirms significant correlations between CRISPR-Cas system types and specific pathogenic traits. Strains carrying type I-F systems exhibited a higher content of virulence genes, while those with type I-E systems showed a greater prevalence of antibiotic resistance determinants. The absence of a CRISPR-Cas system correlated with an elevated prevalence of both virulence and AMR genes. These findings align with the existing literature [14,20] and support the potential of CRISPR-Cas analysis for elucidating mechanisms of bacterial pathogenicity and antimicrobial resistance [21]. Similarly, a higher abundance of plasmid replicons was associated with strains harboring type I-E CRISPR-Cas systems or with strains devoid of any CRISPR-Cas system. This finding is in agreement with a previously published study [14].

We propose that CRISPR-Cas systems can indirectly affect bacterial virulence by targeting and controlling the acquisition of pathogenicity plasmids. Supporting this, we detected spacers matching virulence plasmids characteristic of major *E. coli* pathotypes: a plasmid carrying *ibeC* (associated with meningitis); hybrid plasmids harboring virulence gene sets for EHEC and EIEC/MNEC; a plasmid with ETEC-specific toxins (*eltAB*, *estla*); and a plasmid bearing a large ExPEC-associated virulence arsenal (*iroBCDEN*, *iutA* operon).

## 5. Conclusions

In this study on UPEC strains collected from patients with urological infections in the Central region of Russia in 2004–2019, we have shown the prevalence of UPEC pandemic STs carrying a huge spectrum of virulence and AMR genes. CRISPR-Cas systems were identified in 54% of the strains, including 46% of type I-E and 7% of type I-F CRISPR-Cas systems. Two variants of the type I-E CRISPR-Cas systems were identified based on the differences in *cas* gene sequences. The virulence gene content was higher in strains with type I-F systems, whereas antibiotic resistance genes were more prevalent in strains carrying type I-E systems. Strains lacking a CRISPR-Cas system displayed elevated levels of both virulence and antimicrobial resistance genes. The obtained data expand our understanding of the prevalence of CRISPR-Cas systems among clinically significant UPEC strains and indicate their probable role in this pathogen’s evolution.

## Figures and Tables

**Figure 1 microorganisms-13-02846-f001:**
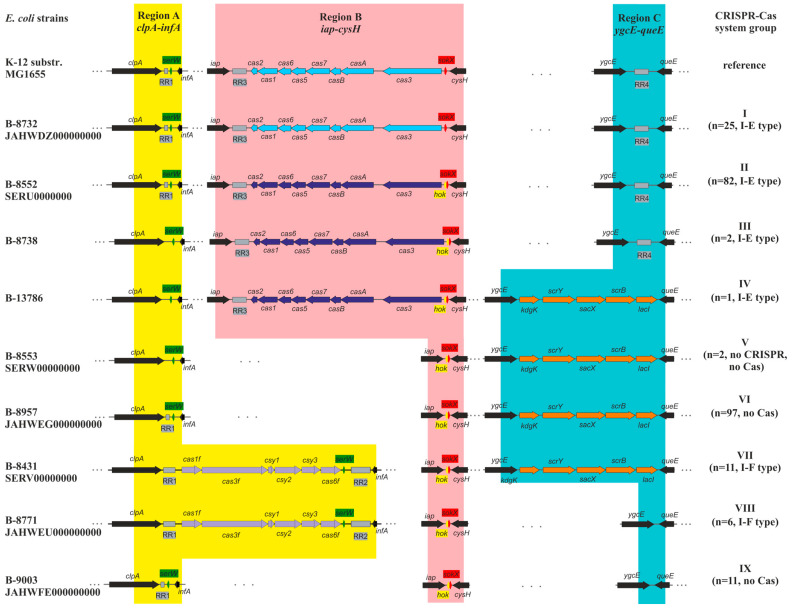
Genomic organization of CRISPR-Cas system components in uropathogenic *E. coli* strains compared to the reference strain K-12 substr. MG1655. Colored areas represent distinct genetic regions: yellow (Region A), pink (Region B), and turquoise (Region C). Gray blocks denote RRs. Genes are indicated by arrows and color-coded as follows: green, *serW* (serine tRNA); red, *hok*/*sokX* (Hok toxin/SokX antitoxin module); light blue, type I-E Cas proteins (K-12 variant); blue, type I-E Cas proteins (B-8552 variant); purple, type I-F Cas proteins; orange, carbohydrate metabolism enzymes; and black, genes unrelated to the CRISPR-Cas system.

**Figure 2 microorganisms-13-02846-f002:**
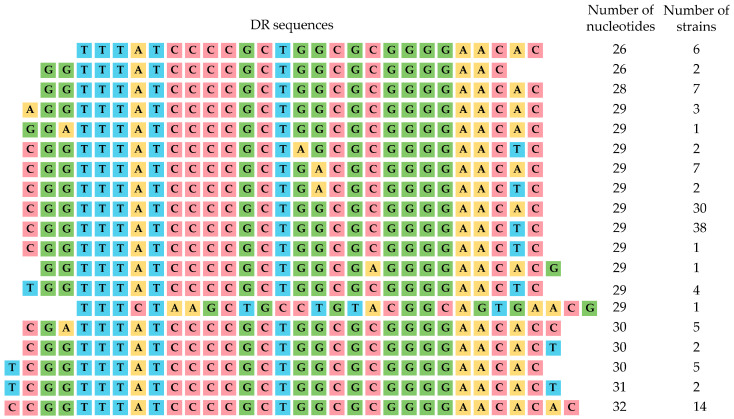
Comparison of DRs located in RR3 and RR4 regions. Nucleotides are highlighted in yellow (adenine, A), blue (thymine, T), green (guanine, G) and red (cytosine, C).

**Figure 3 microorganisms-13-02846-f003:**
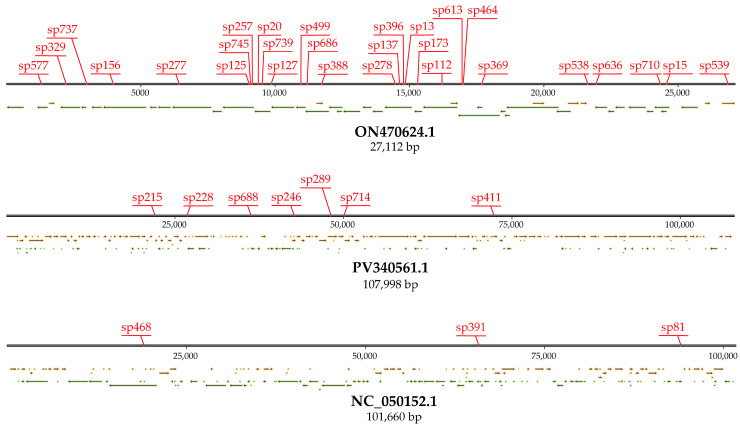
DNA targets for the spacers of the urological *E. coli* CRISPR-Cas systems in genomes of Escherichia phage vB_EcoM-705R4 (ON470624.1), *Escherichia* phage PhiR41_1 (PV340561.1), and *Enterobacteria* phage P7 (NC_050152.1). Red stickers indicate the location of the target sites. Yellow arrows indicate the location of ORFs on the (+) DNA strand, and green arrows on the (-) DNA strand.

**Figure 4 microorganisms-13-02846-f004:**
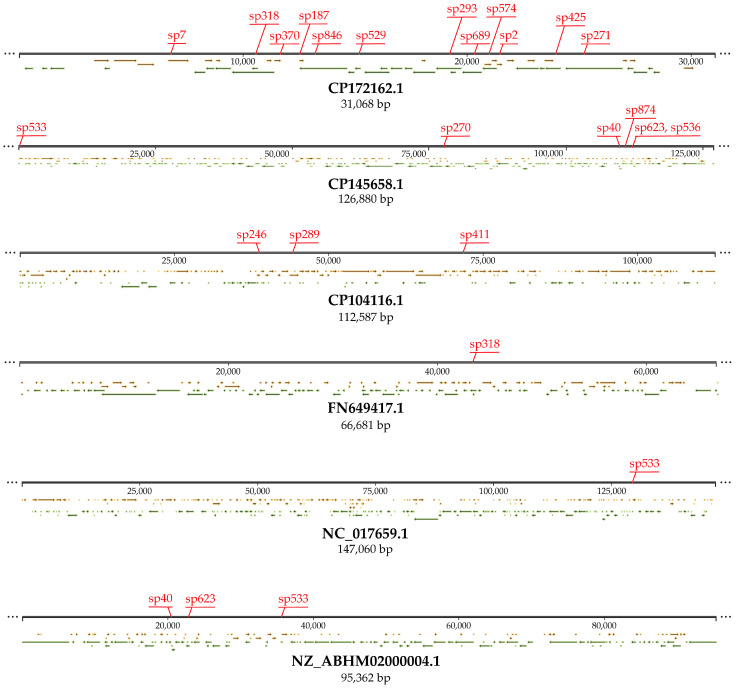
DNA targets for the spacers of the urological *E. coli* CRISPR-Cas systems on the plasmids pGF54-C (CP172162.1), p1-S1-IND-01-A (CP145658.1), pEc2-51408 (CP104116.1), pO157 (NZ_ABHM02000004.1), p666 (FN649417.1), and pO83_CORR (NC_017659.1). Red stickers indicate the location of the target sites. Yellow arrows indicate the location of ORFs on the (+) DNA strand, and green arrows on the (-) DNA strand.

**Figure 5 microorganisms-13-02846-f005:**
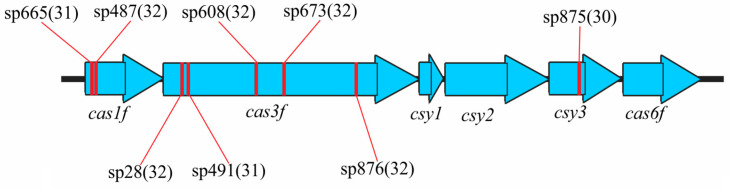
DNA targets for the spacers of the urological *E. coli* CRISPR-Cas systems in the *cas* genes of the type I-F CRISPR-Cas system. Red stickers indicate the location of the target sites. The length of the target DNA sequence in bp is indicated in brackets.

**Figure 6 microorganisms-13-02846-f006:**
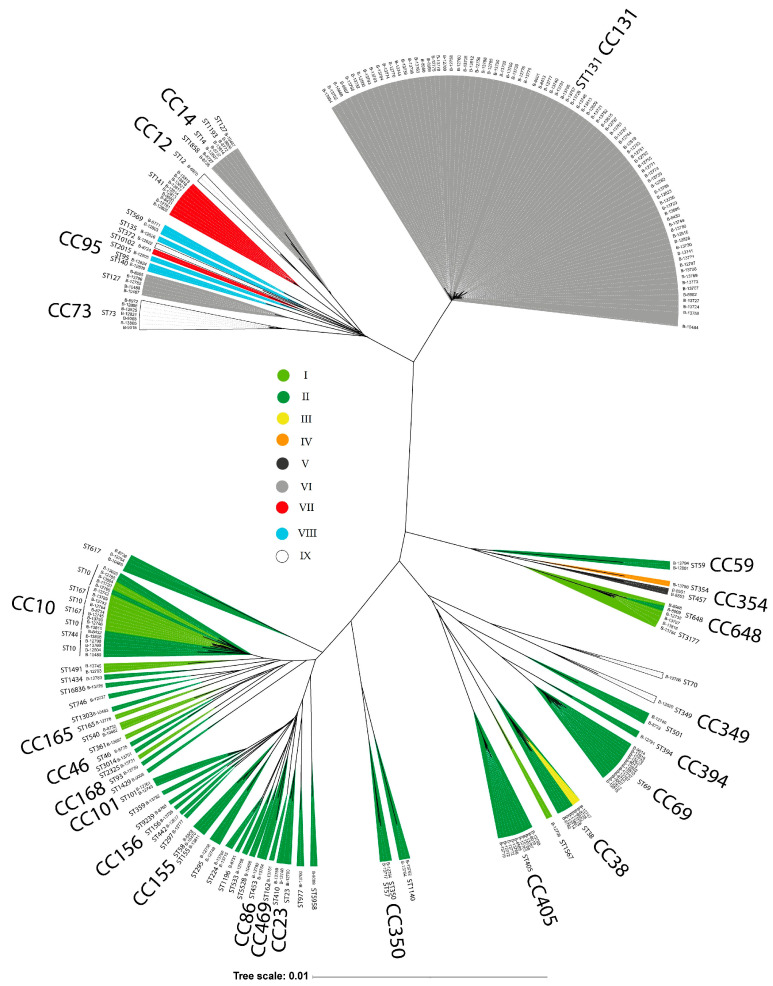
Phylogenetic tree constructed based on whole-genome sequences of urological *E. coli* strains using Snippy version 4.6.0 with default settings and visualized using the iTOL web resource.

**Figure 7 microorganisms-13-02846-f007:**
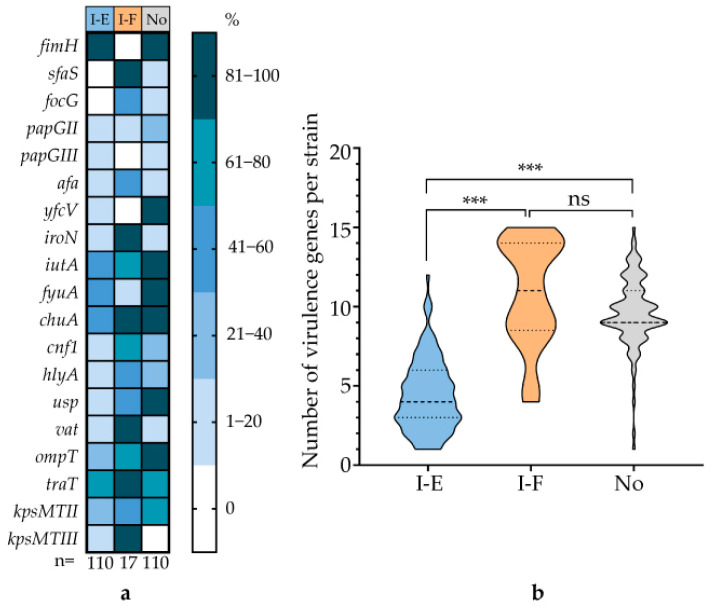
Correlation between CRISPR-Cas system types and virulence gene profiles in UPEC strains. (**a**) Heatmap depicting the prevalence of key UPEC virulence genes, categorized as adhesins (*fimH*, *sfaS*, *focG*, *papGII*, *papGIII*, *afaA*, *yfcV*), siderophores (*iroN*, *iutA*, *fyuA*, *chuA*), toxins (*cnf1*, *hlyA*, *usp*, *vat*), and protectins (*ompT*, *traT*, *kpsMTII*, *kpsMTIII*). (**b**) Statistical comparison of virulence gene content between strains carrying the type I-E systems, type I-F systems, or lacking a system (absence). Significance codes: ***, *p* ≤ 0.001; ns, not significant (*p* > 0.05).

**Figure 8 microorganisms-13-02846-f008:**
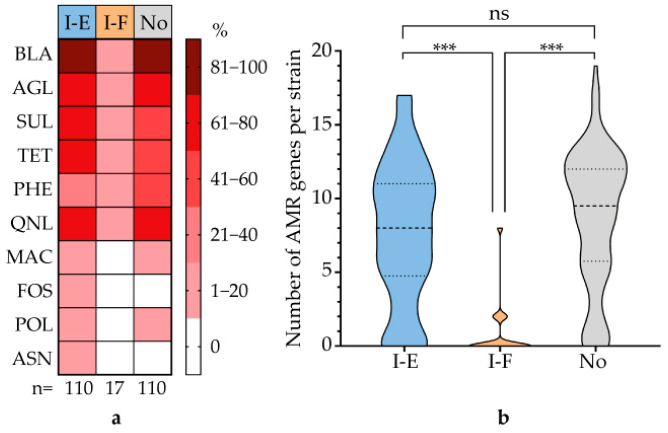
Correlation between CRISPR-Cas system types and AMR gene profiles in UPEC strains. (**a**) Heatmap depicting the prevalence of AMR genes associated with ten functional groups of antimicrobial agents: beta-lactams (BLA), aminoglycosides (AGL), sulfonamides (SUL), tetracyclines (TET), phenicols (PHE), fluoroquinolones (QNL), macrolides (MAC), fosfomycin (FOS), polymyxins (POL), and ansamycins (ANS). (**b**) Statistical comparison of AMR gene content between strains carrying the type I-E and type I-F CRISPR-Cas systems. Significance codes: ***, *p* ≤ 0.001; ns, not significant (*p* > 0.05).

**Figure 9 microorganisms-13-02846-f009:**
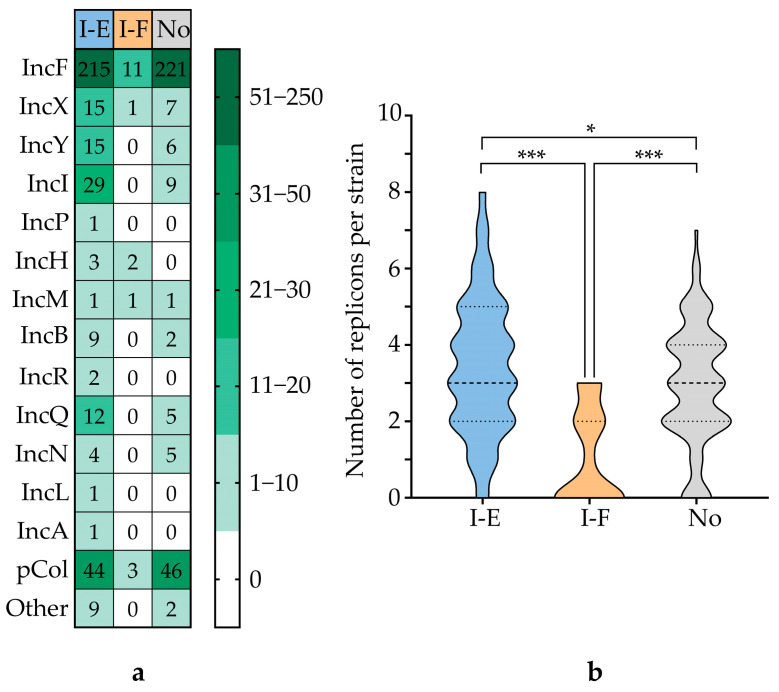
Correlation between CRISPR-Cas system types and plasmid replicons in UPEC strains. (**a**) Heatmap depicting the prevalence of plasmid replicons associated with fifteen Inc groups. (**b**) Statistical comparison of replicon content between strains carrying the type I-E and type I-F CRISPR-Cas systems. Significance codes: *, *p* ≤ 0.05; ***, *p* ≤ 0.001.

**Table 1 microorganisms-13-02846-t001:** Comparison of nucleotide sequences of *cas* genes and amino acid sequences of the corresponding Cas proteins between the var. B-8552 and var. K-12 of the type I-E CRISPR-Cas system in uropathogenic *E. coli* strains.

Gene	var. B-8552/var. K-12			
	Gene Length, bp	GC-Content, %	Gene QC/PI, %	Protein QC/PI, %
*cas3*	2700/2667	50/45	0/0	87/29
*casA*	1563/1509	50/44	1/94	53/23
*casB*	537/483	52/46	0/0	0/0
*cas7*	1056/1092	51/44	0/0	82/31
*cas5*	747/675	55/48	0/0	66/32
*cas6*	651/600	55/45	0/0	99/29
*cas1*	924/918	52/51	90/73	98/84
*cas2*	294/285	48/46	99/75	90/86

Note: QC/PI, Query Coverage/Percent Identity.

## Data Availability

The original contributions presented in this study are included in the article/Appendix A. Further inquiries can be directed to the corresponding author.

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
