# Peer review of "Diversity of CRISPR-Cas Systems Identified in Urological Escherichia coli Strains"

_microorganisms, 2025, doi:10.3390/microorganisms13122846_

Round 1

Reviewer 1 Report

Comments and Suggestions for Authors

The article demonstrates that the CRISPR system is not distributed across all strains. It is a work of considerable scientific rigour and is appropriate for the journal's standards. I only have one question: is it appropriate to include a reprint in this link? doi:10.20944/preprints202510.2396.v1

Comments on the Quality of English Language

The text is easy to understand, but it needs to be reviewed by a native English speaker to ensure excellent quality. 

Author Response

Dear Reviewer 1, we appreciate your kind review of our manuscript. We respond to your comments below.

Comment 1: The article demonstrates that the CRISPR system is not distributed across all strains. It is a work of considerable scientific rigour and is appropriate for the journal's standards. I only have one question: is it appropriate to include a reprint in this link? doi:10.20944/preprints202510.2396.v1

Response 1: We have posted the preprint at this link, in accordance with the Editor's invitation accepted on 28 October 2025.

Comments on the Quality of English Language: The text is easy to understand, but it needs to be reviewed by a native English speaker to ensure excellent quality.

Response to Comments on the Quality of English Language: We thank Reviewer 1 for the suggestion. The manuscript has undergone professional English editing and proofreading to ensure correctness in grammar, technical terminology, and overall clarity suitable for academic publication (MDPI English editing certificate ID: english-103973).

Reviewer 2 Report

Comments and Suggestions for Authors

In the current study, the authors examined the diversity of CRISPR/Cas systems in UPEC E. coli isolates. The findings are of interest, confirming similar results from other studies. The study is organised and presented. 

Comments:

-Didi you observe isolates with CRISPR/Cas systems and without CRISPR/Cas systems belonging to the same ST?

lines 155-194: please explain the impact of this analysis?

lines 221-229: it should be clear that RR3 and RR4 are of type I-E, and RR1 and RR2 of type I-F

lines 266-270: with what regions of the plasmids they matched? These plasmids were associated with resistance or virulence?

lines 313-314: is there a clear conclusion?

lines 305-329: you should perform the same analysis for plasmid replicons.

line 355: correct to Greece.

line 362: cas should be in italics format.

lines 381-409: what about the role of CRISPR/Cas systems in regulation of other functions like virulence? Please discuss it. 

line 393-394: in several previous studies, it has been described that different systems (RR3 and RR4 are of type I-E, and RR1 and RR2 of type I-F) exhibit different DRs.

line 416: did you perform a statistical analysis? If yes, please add the description in the methodology section. Ohterwise delete the specific word.

Author Response

Comments and Suggestions for Authors

In the current study, the authors examined the diversity of CRISPR/Cas systems in UPEC E. coli isolates. The findings are of interest, as they confirm similar results from other studies. The study is organised and presented.

We sincerely thank Reviewer 2 for the time, expertise, and constructive feedback provided. Your comments were very helpful in improving the manuscript, and we have carefully addressed all the suggestions in the revised text.

Comments 1: Did you observe isolates with CRISPR/Cas systems and without CRISPR/Cas systems belonging to the same ST?

Response 1: We thank Reviewer 2 for this question. No isolates with or without CRISPR-Cas systems were found that belonged to the same ST in the studied collection (Figure 6). This information has been added to the section “3.5. E. coli Phylogenetic Groups and CRISPR-Cas Systems Prevalence” (lines 322-323).

Comments 2: lines 155-194: please explain the impact of this analysis?

Response 2: We appreciate Reviewer 2 for this suggestion. The analysis systematically defines the distribution and structural variation of CRISPR-Cas systems within a collection of uropathogenic E. coli strains. By categorizing strains into specific groups based on the presence, type, and genomic organization of these systems. This detailed classification serves as a foundational framework for future studies investigating potential links between specific CRISPR-Cas architectures and bacterial phenotypes. This information has been added to the Discussion section (lines 413-418)

Comments 3: lines 221-229: it should be clear that RR3 and RR4 are of type I-E, and RR1 and RR2 of type I-F

Response 3: We are grateful to Reviewer 2 for this suggestion that prompted us to clarify information in the section “3.4. Analysis of the Repeat Regions, RRs”. Our analysis revealed the following distribution of RRs: in type I-F CRISPR-Cas systems, the combination of RR1 and RR2 was exclusive; in type I-E systems, several combinations were identified (RR1 together with RR3 and RR4, RR3 and RR4 together, or RR3 alone). Additionally, RR1 was found as a standalone repeat locus in strains that lacked any associated cas genes (Figure 1). (lines 225-229)

Comments 4: lines 266-270: with what regions of the plasmids they matched? These plasmids were associated with resistance or virulence?

Response 4: We thank Reviewer 2 for this question. This paragraph has been updated with additional information: Target DNAs for nineteen spacers were identified on plasmids, including targets for twelve spacers (sp2, sp7, sp187, sp271, sp293, sp318, sp370, sp425, sp529, sp574,sp679, and sp846) on the pGF54-C cryptic plasmid of unidentified Inc group (GenBank CP172162.1). Seven spacers (sp40, sp270, sp347, sp533, sp536, sp623, and sp874) matching sequences on the p1-S1-IND-01-A plasmid of IncFIB group (GenBank CP145658.1) carrying both antimicrobial resistance genes (blaCTX-M-15, qnrS1, sul1, aadA5, and dfrA17) and virulence gene ibeC providing the overcoming of the blood-brain barrier and subsequent penetration into the endothelial cells of the brain. Six spacers (sp228, sp246, sp289, sp411, sp688, and sp714) matching sequences on the pEc2-51408 cryptic plasmid of IncFIB group (GenBank CP104116.1). Six spacers (sp40, sp347, sp533, sp536, sp623, and sp874) matching sequences on the pO157 hybrid plasmid of IncFIA:FIB:FII group (GenBank NZ_ABHM02000004.1) carrying a set of virulence genes (hlyABCD, eptO, gspCDEFGHIJKL, stcE, ibeC, and espP) specific for two pathotypes, Enterohemorrhagic E. coli (EHEC) and Enteroinvasive & Meningitis-associated E. coli (EIEC/MNEC). Three spacers (sp536, sp623, and sp874) matching sequences on the p666 plasmid of IncFII group (GenBank FN649417.1) carrying genes eltAB and estIa encoding virulence factors characteristic of enterotoxigenic E. coli (ETEC). Two spacers (sp270 and sp533) matching sequences on the pO83_CORR hybrid plasmid of IncFIB:FIC:QI group (GenBank NC_017659.1) carrying a huge set of antimicrobial resistance genes (blaTEM-1, sul1, sul2, aadA1, dfrA1, tetA, catA1, aph(3’’)-Ib, aph(6)-Id, and mphB), as well as virulence genes (iroBCDEN, iutA, iucABCD, and sitABCD), which are a signature of Extraintestinal Pathogenic E. coli (ExPEC) with high invasive potential. (lines 275-296)

Comments 5: lines 313-314: is there a clear conclusion?

Response 5:

Thank you for your comment. In lines 313-314, we present a clear conclusion regarding the identification of a type I-F CRISPR-Cas system in the strain. The error has been corrected as stated (line 336).

Comments 6: lines 305-329: you should perform the same analysis for plasmid replicons.

Response 6: Thank you for your valuable suggestion. In accordance with your comment, we have performed the analysis for plasmid replicons. The results of this analysis are now presented in Figure 9, and the corresponding description has been added to the manuscript text (lines 361-370). Additionally, we have included an appropriate phrase in the Discussion: Similarly, a higher abundance of plasmid replicons was associated with strains harboring type I-E CRISPR-Cas systems or with strains devoid of any CRISPR-Cas system. This finding is in agreement with a previously published study [14] (lines 448-451). We believe this addition significantly strengthens the manuscript.

Comments 7: line 355: correct to Greece.

Response 7: “USA” has been replaced with “Greece”. (line 388)

Comments 8: line 362: cas should be in italics format.

Response 8: This has been corrected to italic format.

Comments 9: lines 381-409: what about the role of CRISPR/Cas systems in regulation of other functions like virulence? Please discuss it.

Response 9: We propose that CRISPR-Cas systems can indirectly affect bacterial virulence by targeting and controlling the acquisition of pathogenicity plasmids. Supporting this, we detected spacers matching virulence plasmids characteristic of major E. coli pathotypes: a plasmid carrying ibeC (associated with meningitis); hybrid plasmids harboring virulence gene sets for EHEC and EIEC/MNEC; a plasmid with ETEC-specific toxins (eltAB, estla); and a plasmid bearing a large ExPEC-associated virulence arsenal (iroBCDEN, iutA operon). (lines 452-458).

Comments 10: line 393-394: in several previous studies, it has been described that different systems (RR3 and RR4 are of type I-E, and RR1 and RR2 of type I-F) exhibit different DRs.

Response 10:

Indeed, two distinct groups of direct repeats were identified in our study, which is consistent with previous studies [11,30] (lines 429-434). We have modified Figure 2 and added text to the manuscript (lines 238-245).

Comments 11: line 416: did you perform a statistical analysis? If yes, please add the description in the methodology section. Ohterwise delete the specific word.

Response 11: The sentence was modified (lines 466-469).

Reviewer 3 Report

Comments and Suggestions for Authors

The introduction provides an adequate general background. However, it lacks some recent and highly relevant references. It could be expanded to better justify the research gap and the need for the study.

The research design is generally appropriate for the objectives of the study, but certain methodological choices would benefit from clearer justification, particularly regarding sample selection and the use of specific analytical approaches.

The methods are partially described, but don´t provide enough detail for full reproducibility. Some experimental steps, parameters, and software configurations should be specified more precisely.

The results are clearly presented overall. However, some sections are overly concise, and in a few cases results and discussion elements are mixed. Additional clarity and elaboration would strengthen the narrative.

The conclusions are mostly supported by the presented data, but some statements extend beyond what the results can fully address. The authors should adjust the conclusions to avoid overinterpretation and state the study’s limitations.

Comments on the Quality of English Language

Some gramatical mistakes are detected within the manuscript. There are, also, long phrases that could be improved for clarity. It is suggested consistent use of terminology. 

Author Response

We are grateful to Reviewer 3 for carefully reading the manuscript and making comments.

We present the following responses to the comments:

Comments 1: The introduction provides an adequate general background. However, it lacks some recent and highly relevant references. It could be expanded to better justify the research gap and the need for the study.

Response 1. We thank Reviewer 3 for the suggestion. The Introduction section was modified.

Comments 2: The research design is generally appropriate for the objectives of the study, but certain methodological choices would benefit from clearer justification, particularly regarding sample selection and the use of specific analytical approaches.

Response. We thank Reviewer 3 for the suggestion. The Methodology section was completed.

Comments 3: The methods are partially described, but don´t provide enough detail for full reproducibility. Some experimental steps, parameters, and software configurations should be specified more precisely.

Response 3. We thank Reviewer 3 for the suggestion. The Methodology section was completed.

Comments 4: The results are clearly presented overall. However, some sections are overly concise, and in a few cases results and discussion elements are mixed. Additional clarity and elaboration would strengthen the narrative.

Response 4. We thank Reviewer 3 for the suggestion. The Results section was modified.

Comments 5: The conclusions are mostly supported by the presented data, but some statements extend beyond what the results can fully address. The authors should adjust the conclusions to avoid overinterpretation and state the study’s limitations.

Response 5. We thank Reviewer 3 for the suggestion. The Conclusions section was modified.

Comments 6: Some gramatical mistakes are detected within the manuscript. There are, also, long phrases that could be improved for clarity. It is suggested consistent use of terminology.

Response 6. We thank Reviewer 3 for the suggestion. The manuscript has undergone professional English editing and proofreading to ensure correctness in grammar, technical terminology, and overall clarity suitable for academic publication (MDPI English editing certificate ID: english-103973).

Round 2

Reviewer 2 Report

Comments and Suggestions for Authors

The revised manuscript has been signiicantly improved.